# Role of schools in community mobilisation to improve IYCF practices in 6–24-month-old tribal children in the Banswara district, India: findings from the qualitative PANChSHEEEL study

Monica Lakhanpaul [ID] ,[1,2] Susrita Roy,[3] Marie-Carine Lall [ID] ,[4] Hemant Chaturvedi,[5] Rajesh Khanna,[3] Shereen Allaham [ID] ,[6,7] Isabel-Cathérine Demel,[8] Lorna Benton,[1] Virendra Kumar Vijay,[9] Sanjay Sharma,[10] Logan Manikam [ID] ,[6,7] Neha Santwani,[3] Satya Prakash Pattanaik,[5] Tol Singh,[5] Pramod Pandya,[5] Priyanka Dang,[3] Priti Parikh [ID] ,[10] the PANChSHEEEL team

For numbered affiliations see end of article.

**Correspondence to**
Professor Monica Lakhanpaul;
m.lakhanpaul@ucl.ac.uk

## ABSTRACT

**Objective** India has been struggling with infant malnutrition for decades. There is a need to identify suitable platforms for community engagement to promote locally feasible, resource efficient Infant and Young Child Feeding (IYCF) interventions. This study aims to explore if and how schools could represent a site for community engagement in rural India, acting as innovation hubs to foster positive change in partnership with the Angawadi centres.

**Design** Five-phase formative study; A parallel mixed methods approach structured by a socioecological framework was used for data collection at individual, household and community levels. This paper focuses on the qualitative findings.

**Setting** This study was undertaken in nine villages within two blocks, 'Ghatol' and 'Kushalgarh', in the Banswara district of Rajasthan, India.

**Participants** 17 schools were identified. Interviews were conducted with local opinion leaders and representatives in the education sector, including principals, schoolteachers, block and district education officers. Across the nine study villages, information was gathered from 67 mothers, 58 paternal grandmothers using Focus Discussion Groups (FDGs) and 49 key respondents in Key Informant Interviews.

**Results** Schools were considered an important community resource. Challenges included limited parental participation and student absenteeism; however, several drivers and opportunities were identified, which may render schools a suitable intervention delivery site. Enrolment rates were high, with schools and associated staff encouraging parental involvement and student attendance. Existing initiatives, including the mid-day meal, play opportunities and education on health and hygiene, further highlight the potential reliability of schools as a platform for community mobilisation.

**Conclusions** Schools have been shown to be functional platforms frequently visited and trusted by community members. With teachers and children as change agents,

### Strengths and limitations of this study

► The Participatory Approach for Nutrition in Children: Strengthening Health Education Engineering and Environment—HEEE Linkages project was an interdisciplinary study, designed to explore an array of important interconnected themes such as health, education, engineering and environment that influence Infant and Young Child Feeding (IYCF) practices and nutrition in India with the aim of developing a socio-culturally appropriate, tailored, innovative and integrated multisectoral HEEE Nutrition package to optimise IYCF practices for children aged 6–24 months.

► This study established the advantages of using schools to bring communities together and disseminate information regarding IYCF practices while exploring the potential of working with children as community change agents.

► Due to time and resource constraints, the study relied on the coverage and completeness of external data, for example, a list of the children under the age of 2 years resident in the study area (prepared by Anganwadi worker and/or Accredited Social Health Activist), as a starting point for the data collection.

► The study covered only nine villages in total across two blocks in the state of Rajasthan. Despite the main formative study being led across 18 villages, the phase of study directly related to testing the feasibility and acceptability of selected pilot school-based interventions were only conduced in nine villages.

schools could represent a suitable setting for community mobilisation in future wider scale intervention studies. Expanding the supportive environment around schools will be essential to reinforce healthy IYCF practices in the long term.

**Strengths and limitations of this study**

► The data collection phase of this study fell into the months of January to March, the peak of agricultural season in the Banswara district. This impacted our sample size due to absence of functionaries at schools for Key Informant Interviews and Focus Discussion Groups (FDGs) and/or participants in our household survey.

## INTRODUCTION

Many low and middle-income countries (LMIC) have been grappling with the problem of malnutrition for decades. Given its persistent and chronic nature, malnutrition affects a large proportion of the population in these countries. The triple burden of malnutrition presents itself in the form of stunting, wasting and anaemia—all of which are especially common in women and children.

It is well known that the period between 6 and 24 months of age represents a critical time for infant growth.[1] In LMIC, stunting is generally the consequence of infectious disease and low nutrient intake, particularly inadequate energy and protein intake, relative to nutritional requirements.[2] In addition to infectious disease prevention strategies, supplementary Infant and Young Child Feeding (IYCF) interventions targeting this critical period are most effective in reducing malnutrition and promoting childhood growth and development.[1]

In response to this, in 1975, India introduced an 'Integrated Child Development Service' (ICDS)—an intervention comprising supplementary nutrition, immunisation and preschool education to enhance health outcomes for pregnant women, breastfeeding mothers and children up to age 6.[3] The National Programme of Nutritional Support linked to Primary Education (Midday Meal Initiative) delivering hot cooked lunches to all the nation's school children is one of ICDS's flagship interventions, along with the introduction of rural child care centres, so called Angawadi centre (AWC). Linked with schools, the AWCs are aimed at providing services and skilled staff to enhance nutritional outcomes, basic medical aid and preschool education for 2–5 year olds and their mothers in the community setting. While this has led to a significant improvement in the care continuum from pregnancy to early childhood, households from the lowest quintiles of the economic strata and women with poor schooling levels remain difficult to reach.[2] Identifying occasions and settings where community members (CM) gather at regular intervals with children will, hence, be crucial to increase the reach of similar interventions going forward, especially in rural states.

Schools appear to be one potential setting for community mobilisation and implementation of interventions, as they offer continuous, intensive contact with children. By contributing to students' health literacy and behaviours, school-based health promotion can be particularly valuable in the context of developing countries, where low public health literacy and high burden of disease are faced.[4] Furthermore, children have been demonstrated to be change agents who are able to transfer information and play as an active partner in the school–home relation.[5] In a literature review on school-based health promotion and their impact in developing countries, it was found that a majority of school-based interventions addressed nutrition, hygiene, oral health and communicable diseases,[4] through nutrition education and hygiene awareness campaigns. Implementation was carried out by schoolteachers, nurses and other CM, with study results suggesting an improvement in the knowledge and attitudes of their target population.[4 5]

In an attempt to maximise the potential of schools and achieve greater convergence, and an improved continuum of care for early childhood, the Government of India (GoI) subsequently scaled up the so-called 'Rajasthan model'—an initiative that leads to the establishment and colocation of 11 000 AWCs with schools by mid-2017 in rural India.[6]

This soon led to schools emerging as a logical platform for intervention delivery in the context of the Participatory Approach for Nutrition in Children: Strengthening Health Education Engineering and Environment Linkages (PANChSHEEL) project. At the conceptualisation stage, schools, in conjunction with the government-funded, colocated AWCs, were, thus, considered as a potential platform and resource for community mobilisation to improve IYCF practices in rural India.

In this paper, we present the qualitative findings from our formative study, exploring the acceptability and feasibility of schools as a platform for community mobilisation and health promotion activities related to IYCF practices and the role of children as community change agents.

## METHODOLOGY

### Study design

This present study forms part of a wider project: The PANChSHEEL—initiative; a project involving a five-phase formative interdisciplinary study, designed to explore health, education and environmental (HEEE) factors that influence IYCF practices and nutrition in India. Informed by the socioecological model, the study aimed to develop a socioculturally appropriate, tailored, innovative and integrated cross-sector HEEE Nutrition package to support optimal IYCF practices for children in rural India aged 6–24 months.[7–9]

In this project, members from University College London, Institute of Child Health, Institute of Education and Faculty of Engineering Sciences and expertise from; Save the Children-India, Indian Institute of Technology, New Delhi and Jawaharlal Nehru University, New Delhi formed the core research team (CT). The participatory approach actively involved local community researchers (CR), community champions (CC) and CM selected from the nine villages within Ghatol and Kushalgarh Block of Banswara district of Rajasthan (India).

The study was conducted in tribal areas including households from the lowest quintiles of the economic strata and women with poor schooling levels. At the outset of the study, understanding and reflecting on the village structure and positioning of schools in relation to other village resources were key for the field team. Village profiles were, therefore, created by the means of social mapping and community profiling. Social mapping was conducted using a participatory approach, with 10–15 participants per village being identified by key figures in the community (eg, elected village representative, school teachers and other local leaders). Field investigators acted as facilitators, explaining the purpose of drawing a village map and working with the participants to produce it. The first step in this process was drawing the concrete or pucca roads, followed by the foot-roads or kuccha roads. Important institutions (eg, temples, government schools), houses and water sources (eg, hand pumps, canals) were then marked on the map. Participants were required to discuss with one another before reaching a consensus on the final map. The participants' map was then validated by the village representatives.[9]

Phase 1 of the study focused on identifying and documenting local practices with regards to IYCF practices, sanitation, education and access to local resources (such as energy and water). The existing GoI initiatives in the study sites were also mapped during phase 1. Phase 2 involved extensive fieldwork to identify local challenges, drivers, resources, opportunities and needs for children in the first 6–24 months of life at individual, household, community and environmental levels. Phases 3, 4 and 5 focused on the analysis and mapping of HEEE linkages, the cocreation and piloting of an IYCF intervention package (IP) using an iterative process as well as review and dissemination phases of this project.[10] This present paper focuses only on the qualitative findings relating to the potential role of schools in community mobilisation to support improvements in IYCF from the formative phase (phase 1). The full study process is presented in two available reports.[11]

### Schools selection
The formative research phase commenced after a systematic selection of nine study villages from the two blocks of Banswara, namely, Ghatol and Kushalgarh. Out of the 18 villages proposed initially by 'Save the Children India', 9 were selected purposively based on an interaction of the CT with (1) the local community, (2) the teachers at the local elementary schools and (3) the Anganwadi workers (AWW) and Accredited Social Health Activist (ASHA) in each village. Schools where the relationship with the local village and functionality of the school itself were poor, eg, low number of teachers and students, insufficient engagement with the study, were excluded. Second, villages where a lack of interaction between different social groups or lack of interest in the project was noted, were subject to exclusion. Finally, the linkage of the schools with the AWCs, for example, the

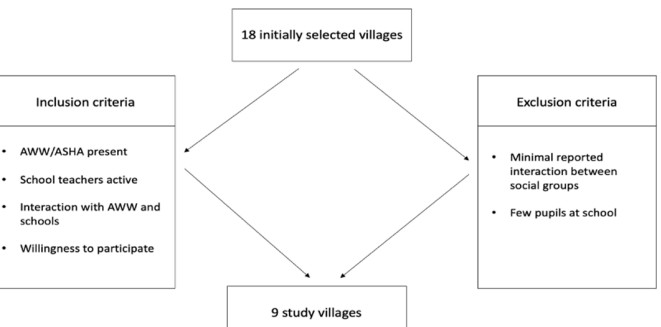

**Figure 1** Inclusion and exclusion criteria for the study villages and schools. ASHA, Accredited Social Health Activist; AWW, anganwadi workers.

government-funded centres that provide facilities such as food, vaccines, medical treatments to 2–5-year olds alongside assisting mothers in accessing other government schemes/services, was a key determining factor. Schools, where the AWC was not directly associated with the school or where the AWW displayed insufficient interaction, were excluded. A summary of the inclusion and exclusion criteria for the study villages and schools is outlined in figure 1. Overall, 17 schools were identified in the nine selected villages.

### Patients and public involvement and engagement
Patient/participant and Public Involvement and Engagement representatives were identified, selected and engaged in every step of the study from protocol development over study design to the dissemination of the results.

CM from the nine villages within Ghatol and Kushalgarh Blocks of the Banswara district of Rajasthan were actively supporting the CT in the context of PANChSHEEEL's participatory approach. To help steer the project across its phases locally, CRs from the study blocks Ghatol and Kushalgarh were recruited. The inclusion criteria encompassed (1) being educated, (2) having previous field experience in Banswara district, for example, understanding the village context and social networks locally and (3) being conversant in both 'Hindi' and 'Wagdi' languages (given that the most prevalent language dialect in the nine study villages is 'Wagdi'). With the help from the 'Save the Children' Team from the Rajasthan State Office, two suitable individuals were identified. The CRs were initially male adult members of the study population who understood the local village and social context, providing special insight about their communities and environment to the study. Both had shown satisfactory engagement in a previous project by 'Save the Children', were local to the study blocks of Ghatol and Kushalgarh and, hence, recruited on contractual basis during phase 1 of the study to support community engagement and data collection. Their main role was to identify potential candidates who could engage actively in project activities as well as engaging with the CCs, who in turn helped establish linkages with the CMs and improve networks between schools, frontline workers (FLWs) and villages.

Aligning with the principle of reflexivity in research, that is, the continuous process of examining relationships, the dynamics of that relationship and its impact,[11] two female researchers were subsequently recruited in addition (for short term). This decision was made in response to the situational dynamics in our study setting[11] and feedback from local CM (such as teachers and AWW), who felt that the female respondents would discuss sensitive topics of breastfeeding and complementary feeding more freely with a woman, rather than men, CR in the quantitative household survey. The following criteria were, hence, set for recruitment purposes: the chosen female CR should (1) be a graduate, (2) have experience in interviewing female CM (3) be familiar with the local dialect, that is, Wagdi and (4) be prepared to travel/commute within the study district.

CCs were associated voluntarily in the research. They played no direct part in data collection but did have an important role in being advocates for the study and facilitating access to the community themselves.

### Data collection

For the whole PANChSHEEEL study, a mixed-methods approach structured by a socioecological framework was employed for data collection. Primary data were collected at an individual, household and community level by the CT and CR together. Qualitative data were obtained using two methods: (1) Key Informant Interviews (KII) with AWWs, ASHAs, Auxiliary nurse Midwifes (ANMs), Elected Panchayat Representatives (PRI members), school teachers and principals, relevant block and district officers and (2) Focus Group Discussions (FGD) with mothers and grandmothers of children 0–2 years of age. While not directly involved in the data analysis, findings were presented to CRs at each stage for verification and interpretation purposes.

Participants were asked for consent in the context of both the qualitative and quantitative data collection. Written consent was obtained from participants of the KIIs. Verbal consent was obtained from all participants of FGDs and recorded by the means of an audio device. All interviews were documented with a voice recorder for subsequent analysis. Field notes were taken during the interviews with stakeholders. The coding was carried out by the field team and was done first in Hindi and, subsequently, in English.

The members of the research team conducting the qualitative data collection received a 2-day training by an experienced coinvestigator in the Banswara district in December 2017. This training was followed by a 2-day field practice. Data were collected by the CRs with assistance from CCs for scheduling meetings with stakeholders. The CR were supported by the CT and received thorough training on techniques and tools to FGDs, KII and broad narrative group discussions. PowerPoint presentations and handouts were used for training and reflective sessions were organised with the CT.

During the formative phase and stakeholder mapping workshop, relevant stakeholders were identified and recruited. Considering the possibility of snowballing, a sampling method where existing participants inform the recruitment of future participants from among their acquaintances and relevant to the study, the research team was open to any new stakeholder(s) identified during the data collection.

The CMs were interviewed in FGDs, whereas key informants (KII) were interviewed individually. A focus was placed on interviewing paternal grandmothers, primarily due to their significant influence on mothers: after the delivery, it is common for mother and infant to spend the initial postpartum and puerperal period in her husband's living environment. This makes paternal grandmothers key in guiding the mother after the delivery on breastfeeding and complementary feeding practices. All participants in the mothers' group had children below the age of 2 years; most of them also had an older school-going child. Similarly, all grandmothers who had a grandchild under 2 years of age were included. Interviews were also held with other key informants from the village namely elected representatives, ANM, AWW and ASHA, and the village school teachers and principals. Block and district level officers from the Health, Education and ICDS departments were also interviewed. The recruitment of participants was conducted based on availability and willingness to participate.

### Topic guide

The CT and CR designed thematic guides designed through literature review, expert and stakeholder consultation and evolved through pilot testing with different population subsets.

### Data management and analyses

Qualitative data were collected in Hindi or Wagdi and transcribed and translated by local translators who were proficient in Wagdi and Hindi. Research team members who were proficient in Hindi and English translated the Hindi texts to English. At each stage, quality assurance measures were adopted by checking for reverse translation. Coding was done by two junior researchers who had considerable experience of the study sites and contexts. Final consensus was arrived at collectively by the senior researchers of the team. Data were collected from schools in all the nine study villages.

The qualitative data collected were transferred from the recording device to the encrypted laptop of the Save the Children India (SCI) Nodal Person. The research team also met at the end of each day to share the key observations and reflections about the data based on the notes taken during the KII and FGD.

Audio-recorded data were translated and transcribed alongside detailed summary of field notes prepared. Data from open-ended questionnaire questions were evaluated, discussed and summarised across the various IYCF themes for each intervention point. The coding was

**Table 1** Summary of interviewers (numbers by participant group and type of interaction) across the study villages of Ghatol and Kushalgarh

| Participant group | Type of interaction (FGD/KII) | N (Ghatol—five study villages) | N (Kushalgarh—four study villages) | N (total across the nine study villages) |
|---|---|---|---|---|
| ANM | KII | 4 | 3 | 7 |
| ASHA | KII | 7 | 6 | 13 |
| AWW | KII | 7 | 6 | 13 |
| School teacher | KII | 5 | 4 | 9 |
| Sarpanch/ward panch | KII | 4 | 3 | 7 |
| Mothers | FDG | 36 | 31 | 67 |
| Grandmothers | FDG | 30 | 28 | 58 |

ASHA, Accredited Social Health Activist; AWW, Anganwadi workers; FDG, focus group discussions; KII, Key Informant Interviews.

conducted by the field team and was done first in Hindi, and subsequently, in English.

The initial data collected by CR further underwent respondent validation. Respondent validation was conducted in a workshop involving CCs facilitated by CR under the guidance of SCI staff. The steps of a thematic analysis were employed; this means identifying, analysing and interpreting patterns of meaning (a.k.a. themes) within the qualitative data sets.[12] The qualitative data analysis was analysed by free listing and then subcategorised based on emergent themes.

## RESULTS

For the purpose of this paper, we have narrowed down our findings to those relevant to schools within the two blocks.

Across the nine study villages, information was gathered from 67 mothers, 58 paternal grandmothers using FDGs and 49 key respondents in the KIIs. Table 1 illustrates a summary of the interviewers across the study villages of Ghatol and Kushalgarh.

Findings from both KIIs and FGDs revealed several activities and facilities provided by schools that may render them a suitable site for community mobilisation. The challenges, drivers and opportunities of schools as change agents across the nine examined study villages are summarised in figure 2.

### Challenges
#### Parental participation in schools
Across a majority of schools studied, parent–teacher interaction was reported to be positive, but not optimal. Parents generally reported attending two kinds of school events: (1) national day celebrations in school (eg, independence day or republic day) or (2) if a parent was summoned to discuss their child's progress. School management committee meetings were also held in the school once per month. Both teachers and mothers, from the study village, were of the opinion that parents usually visited formal school meetings and some teachers reported organising meetings at the community level.

While not offered at all schools, the teachers conducting such community meetings felt that they were able to effectively engage with the community. However, it was difficult to maintain the regularity of such events due to teacher shortages.

> We go inside the village to interact with the parents. The School Management Committee helps to fix a date, time and place for these meetings. These meetings are useful. […] It is easy for them to attend those meetings, but we cannot do it frequently (school teacher, Ghatol).

### School attendance
Teachers and mothers reported school attendance to be high throughout the year. Absenteeism was identified as a problem mainly during harvesting season by teachers in both blocks, specifically in Ghatol where school-aged girls were kept at home to take care of younger siblings while boys assisted the family in the agricultural work.

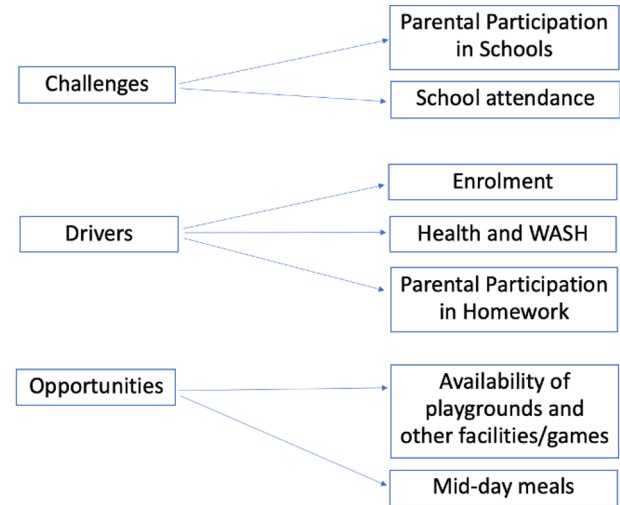

**Figure 2** Overarching themes identified during the thematic analysis of KIIs and FGDs: challenges, drivers and opportunities. WASH, Water Sanitation and Hygiene.

Children mostly miss school when there is harvesting season. Sometimes the entire family gets busy so there is no one to send the child to school. The girls also miss school when they have to take care of their little siblings, but rarely (school teacher, Ghatol).

In cases where students were absent for long stretches of time, it was found that teachers conducted home visits to explore the reason for the non-attendance, hence playing an important role as change agents.

## Drivers
### Enrolment
Evidence from the interviews showed that school enrolment was conducted by the means of door-to-door identification by the primary school teachers. All children above 5 years of age were identified and validated with birth listing done by Frontline Health Workers like AWW and ASHAs. The teachers across the nine villages reported this method to be successful, allowing a recruitment rate of 100%, with no children under 5 years not attending school.

There are no children who are not in school. All children above 5 years come to school (school teacher, Kushalgarh).

In Ghatol, it was reported that some children did not attend governmental school given the existence of a private school in the vicinity of three out of five study villages. This was, however, not the case for Kushalgarh where none of the study villages had access to private schools and all enrolled students attended government schools.

School enrolment is not a problem in this village. Once a year we do a door to door survey to identify school going children. Some families who can afford to send their children to the private do so, rest send them to the government school (school teacher, Ghatol).

High school enrolment levels across the nine villages studied indicate the potential reliability of schools as an intervention delivery platform. Furthermore, the regularity with which children were found to attend school and the degree of parental participation in school activities are important considerations.

### Parental participation in homework
According to the teachers from both blocks, homework was given regularly to reinforce the lessons taught. The response to and success in completing the tasks correctly, however, varied across children in the two study blocks and can broadly be divided into three categories: (1) children who failed to complete the tasks entirely, (2) children who completed the tasks, but with some errors and (3) children who completed the tasks with no errors. The teachers mentioned in the interviews that they believe this difference in performance to be at least partially attributable to the level of engagement of the children's family members.

Only educated mothers can help. We just tell the child to study (mothers, Ghatol and Kushalgarh).

The first category of students was mostly those who stayed with their grandparents or the parents who were illiterate and, therefore, had little to no support for in their education.

Most children stay with the grandparents. They do not have any environment for studying at home. Homework is done at schools. (school teacher, Kushalgarh).

The second category of students were those where the child's mother was educated till fifth standard or had a school going elder sibling who assisted him/her. Occasionally, the parents of children in the third category were engaged in teaching the child, hence leading to better performance in homework tasks. According to teachers in Kushalgarh, the majority of students belonged to the first category and the remaining few were in the second category. In case of Ghatol, the first category of students was relatively smaller. In general, these results suggest that teachers are an important reference point for students and play an essential role in supporting students, especially where limited support is available at home.

### Health and Water, Sanitation and Hygiene (WASH)
The school adopted different approaches to improve student health and hygiene practices. For health, an annual check-up was conducted each year by ANMs. The teachers explained that ANMs came with adequate medicine for camps, and mostly provided first aid treatment. They also identified children who needed special medical attention. Health services were also available from their local AWC. The teachers felt this was a successful and necessary initiative, given the generally limited knowledge about health and hygiene in the local community.

Hand pumps available in the premises of most schools were used as a drinking water source though the water quality was not tested regularly. In a majority of schools, the same hand pumps also used for handwashing. In a few schools, special filtering plants were present. Toilets were available for children, in schools; while only few had separate facilities for boys and girls. Overall, the toilets in most schools were in an unsatisfactory condition, with a large proportion of toilets dysfunctional.

Several teachers reported the provision of special programmes related to diarrhoea prevention and hygiene promotion in schools. In these programmes, awareness was generated about health and hygiene practices. The teachers felt that children proved to be an effective channel to pass on this information at home, hence fostering better understanding about nutrition, health and hygiene practices, also across generations.

We tell the students about good hygiene practices like handwashing, taking a bath every day, cutting nails. They then go and tell their family. Spreading messages through children are very effective (school teacher, Ghatol).

## Opportunities
### Availability of playgrounds and other facilities/games
The presence of playgrounds or outdoor activity areas was noted among all except two schools visited. Additionally, physical education teachers and sporting facilities were available in a few schools. It was reported that facilities were mostly only operational during summer rather than rain season. In the KIIs, several teachers noted that they incorporate play activities in the last period of every school day, since no such opportunities were available at the children's homes. The teachers believe this to be essential for the development of the children, promoting both their physical and mental well-being.

Playing is very necessary for the holistic development of children. We devote the last period each day for that (school teacher, Ghatol).

FGDs showed that not all parents were convinced of the usefulness of play activities for their children.

Parents sometime question us why we play games, school is for studying not playing (school teachers, Kushalgarh).

This highlights another potential area where teachers could use their positive standing in the local community to influence and change current practices.

### Mid-day meals
All teachers, PRI members and mothers responded that the mid-day meal was prepared in the school and served regularly to students. The menu for each day was fixed by the education department, with diet plans including cereals such as wheat and rice, pulses, vegetables. Fruit was also served once a week. The food was prepared by a cook and an assistant under the supervision of the principal or other teachers.

I have seen food being served in schools regularly. Sometimes I go to check the quality also (Ward Panch, Ghatol).

No concerns regarding school meal quality were raised in the FGDs with mothers, nor in the interviews with PRIs. The teachers, especially from schools in Kushalgarh, were of the opinion that nutrition diversity offered in mid-day meals was superior to foods served to children at their homes.

Yes, we provide food to children regularly. [...] The food that they get in schools is better than what they get in home. Dal is not cooked in the homes every day. Fruit is also an attraction among kids (school teachers, Kushalgarh).

Furthermore, it was reported that the general standards of cleanliness and hygiene, for example, when storing, preparing and serving food, were higher at the schools than those encountered at home: at schools, clean water was always used to prepare and cook meals and hygiene practices were implanted for hands, cooking equipment and cutlery.

## DISCUSSION
Appropriate supplementary IYCFs are inherently linked with growth and development, especially in children younger than 24 months. Based on the evidence globally, universal coverage of optimal breastfeeding could prevent 13% of global deaths occurring in children less than 5 years of age, while appropriate IYCF practices could result in an additional 6% reduction in under-5 mortality.[1]

Efforts to improve IYCF hence remain a potent intervention target. Awareness generation among mothers and families with young children is crucial for successful intervention implementation. Identifying occasions and settings where CMs gather at regular intervals, hence is a crucial step to increase reach of possible health interventions, especially in rural states.

Since 1986, the WHO has placed particular emphasis on the potential of health promotion strategies[13] in a school-based setting, but the sustainability of school-based interventions has remained relatively unexplored, especially in LMIC.[14] Traditionally, the role of schools is to build human capital and social cohesion. During the era of millenium development goals (nowadays known as 'sustainable development goals'), school-enrolment and universal primary education have increased in developing countries, giving school-based interventions, the potential to reach significant numbers of children.[5] School infrastructure and physical environment, hygiene campaigns, policies, curricula and teachers have the potential to positively influence physical, sociocultural and policy characteristics of the microenvironment.[15] Moreover, schools are well known to be used as a centre of change especially during crises in high-income countries[16 17] as well as LMIC.[18 19]

In our study, schools have been shown to be an important and viable setting for community mobilisation in the context of nutrition, health and hygiene practices. The qualitative evidence from the FDGs and KIIs, as well as intervention development and the piloting of the fore-mentioned interventions, indicates that schools may indeed represent a feasible potential platform for community engagement and mobilisation. Many existing, local government bodies within the village, for example, AWCs, Health Sub Centre, Panchayat, Public distribution system, are closely associated with the schools, again rendering them a potentially suitable site for intervention delivery to collaboratively improve IYCF practices.

Enrolment rates were high across the nine villages investigated in this study. Schools and associated staff were dedicated to encouraging parent involvement in the

school and discouraging student absenteeism. Initiatives, such as the mid-day meal, play opportunities and education on health and hygiene (eg, through the WASH), all highlight the schools' potential for community engagement and reliability as an intervention delivery platform. Festivals and local celebrations such as Janmashtami, Independence Day, Republic Day, Annual Day Celebration take place on the school campuses. Furthermore, schools can act as research sites, playing a crucial role in capturing the diversity and dynamics that exist in particular communities. As one of the only functional places in the rural communities, they represent the 'beating heart' of villages, where CMs of all generations gather at frequent intervals.

In this context, teachers were found to be a crucial stakeholder in bringing about positive change in local communities, acting as a reliable and sophisticated source of information regarding the communities they serve alongside being potential change agents for children, their parents and the wider community via; (a) their connections and respect from their local communities and (b) their interest in codesigning interventions with us. Thus, engaging with the schoolteachers in further analysis of the problem and codesigning any future interventions holds merit. This has been recognised locally and was translated into initiatives of the District Collector of Bansawar, including the Swachhta Pakhwada and Alakh programmes where school staff, parents and PRI members gather at regular intervals to discuss challenges and improvement opportunities for the village.

Furthermore, many school-based interventions have the advantage of being a cost-effective method to deliver interventions with a broad reach, enrolling participants from different socioeconomic backgrounds, ages and at-risk groups.[19] It is evident from the literature that children have acted as agents of change in multiple health-related community interventions to improve their own and their parents' health.[20 21] Studies have demonstrated that children learn about a health topic and share that information with their parents or their community, thereby fostering an increase in health awareness and health-promoting behaviours across generations.[22 23] This combined with children's quick learning ability and regular access to high-quality information through schools and associated staff gives children the potential to become positive community influencers.

Finally, while the findings from our study strongly suggest that schools represent a suitable site for community mobilisation and health promotion in rural India, the possible limitations of our primary data collection need to be considered. One significant challenge was that of time—for efficiency purposes, data collection was restricted to 2 hours per village; hence, possibly limiting the information gathered from the study participants in the FDGs and KIIs. Second, conducted between January and March 2018, the timing of our primary data collection exercises overlapped with the peak of agricultural season in the Banswara district. This may have impacted our sample size, due to absence of functionaries at schools and/or preoccupation of participants of our household survey with agricultural activities.

In the future, efforts, thus, need to be focused into gathering data from a larger sample size and area, to capture the complexity of actual practices and, hence, gain a more holistic overview of current challenges and opportunities. Our qualitative findings will then be able to translated into practice, namely, by developing culturally sensitive, locally effective, school-based interventions to positively influence in IYCF in rural India. Future studies using the PANChSHEEEL model (IYCF IP combined with multilevel engagement with schools and AWC at the centre) will investigate this further and explore how the colocation with the Angawandi centres may be best leveraged in subsequent intervention development and implementation.

## CONCLUSION

Schools are likely to act as a central theme in deployment for behaviour change interventions, owing to the fact that they form a central part of the child's daily routine in addition to schools being a centre of community activities. These activities could be incorporated into the ongoing programmes, through either leveraging of the mid-day meal programme or other such child development schemes that will provide opportunities to use these interactions to improve the wider determinants of healthy living in children as well as their families.

Developing a supportive environment around schools, geared towards reinforcing messaging on healthy practices and using children as change agents to bring about community change, therefore, requires further investigation.

**Author affiliations**
[1]Population, Policy and Practice, UCL Great Ormond Street Institute of Child Health, University College London, London, UK
[2]Whittington Health NHS Trust, London, UK
[3]National Support Office, Save The Children, Gurugram, India
[4]Institute of Education, University College London, London, UK
[5]Save the Children, Rajasthan, India
[6]Department of Epidemiology and Public Health, University College London Institute of Epidemiology and Health Care, London, UK
[7]Aceso Global Health Consultants Ltd, London, UK
[8]Department of Life Sciences & Medicine, King's College London GKT School of Medical Education, London, UK
[9]Indian Institute of Technology Delhi (IIT), New Delhi, India
[10]Engineering for International Development Centre, University College London, London, UK

**Acknowledgements** The opinions in this paper solely reflect the views of the authors, not of organisations who financially supported this research. All data used in this research is the property of the authors. It is only obtaineable by email from the corresponding author.

**Collaborators** On behalf of the PANChSHEEEL team (Professor Monica Lakhanpaul, Susrita Roy, Dr Lorna Benton, Professor Marie Lall, Rajesh Khanna, Professor Virendra Kumar Vijay, Sanjay Sharma, Dr Logan Manikam, Neha Santwani, Dr Hanimi Reddy, Hemant Chaturvedi, Shereen Allaham, Satya Prakash Pattanaik, Tol Singh, Pramod Pandya, Priyanka Dang, Dr Priti Parikh), we gratefully acknowledge the input from our community champions and field team which made this project

possible. We would like to thank our research partners; Madhav Foundation (Ritu Chhabria & Ritu Prakash), EKAM Foundation (Dr.Sai Lakshmi, Ms. Neeta Karal Nair, Mr. Manoharan, Dr. Sathya Jegannathan and Ms. Benita), Foundation for Research in Community Health India (Dr Nerges Mistry & Dr. Shilpa Karvande), Society for Nutrition Education & Health Action (SNEHA; Dr Nayreen Daruwalla and Dr David Osrin) and our technical advisory board (Prof Anita Saxena, Mr Anand Karve, Prof V M Chariar, Prof T. Sundararaman, Dr Ian Warwick, Dr Ramesh Mehta, Prof Atul Singhal, John Pelton, Professor Sachin Maheshwari, Dr Sofia Strummer, Mr Himanshu Parikh & Dr Amita Kashyap) for their input throughout the study.

**Contributors** ML, ML, PriP, RK, and SS conceived the original concept of the study and designed the research methodology. SR, ML, PraP, RK, HC, SS, SPP, TS and PriP carried out the interviews, analysed the data and contributed to manuscript writing. ML, LM, PriP, SR, HC, NS, SS, SPP, LB, VV, PD and RK, interpreted the data, validated the study and revised the manuscript critically for important intellectual content. SA and ICD wrote the manuscript, contributed to the data analysis, edited the final manuscript and prepared for submission. ML had primary responsibility for the final editing of the manuscript and content, and responsible for the overall content as the guarantor. All authors read and contributed to reviewing the analysis of the data, the designing of the manuscript, and the approval of the final manuscript.

**Funding** This research was supported by the MRC Global Challenges Research Fund [grant number: MR/P024114/1]. Prof Monica Lakhanpaul was supported by the National Institute for Health Research (NIHR) Collaboration for Leadership in Applied Health Research and Care (CLAHRC) North Thames at Bart's Health NHS Trust, and was supported by the NIHR Biomedical Research Centre based at UCL Great Ormond Street Institute of Child Health/Great Ormond Street Hospital NHS Foundation Trust. Dr. Priti Parikh is currently supported by the BBOXX/Royal Academy of Engineering Senior Research Fellowship.

**Competing interests** None declared.

**Patient and public involvement** Patients and/or the public were involved in the design, or conduct, or reporting, or dissemination plans of this research. Refer to the Methods section for further details.

**Patient consent for publication** Not applicable.

**Ethics approval** The PANChSHEEEL Project protocol, including qualitative and quantitative data collection methods as well as the tools obtained ethical approval from the UCL ethics committee (Ref number: 4032/002) in UK and the Sigma-IRB (Ref number: 10025/IRB/D/17-18 Ref number: 4032/002) in India.

**Provenance and peer review** Not commissioned; externally peer reviewed.

**Data availability statement** Data are available upon reasonable request. The data of this study are available from the corresponding author upon reasonable request.

**ORCID iDs**
Monica Lakhanpaul http://orcid.org/0000-0001-5288-3325
Marie-Carine Lall http://orcid.org/0000-0002-2868-4534
Shereen Allaham http://orcid.org/0000-0003-0275-3228
Logan Manikam http://orcid.org/0000-0001-5288-3325
Priti Parikh http://orcid.org/0000-0002-1086-4190

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
