## [Reviewer comments · BMJ Open]

ARTICLE DETAILS

TITLE (PROVISIONAL)	The Role of Schools in community mobilisation to improve IYCF practices in 6-24 month old Tribal children in the Banswara district, India – Findings from the qualitative PANChSHEEEL study
AUTHORS	Lakhanpaul, Monica; Roy, Susrita; Lall, Marie; Chaturvedi, Hemant; Khanna, Rajesh; Allaham, Shereen; Demel, Isabel-Cathérine; Benton, Lorna; Vijay, Virendra; Sharma, Sanjay; Manikam, Logan; Santwani, Neha; Pattanaik, Satya Prakash; Singh, Tol; Pandya, Pramod; Dang, Priyanka; Parikh, Priti

VERSION 1 – REVIEW

REVIEWER	Nakawala Lufumpa University of Birmingham College of Medical and Dental Sciences, Institute of Applied Health Research
REVIEW RETURNED	03-Feb-2021

GENERAL COMMENTS	Abstract It is a bit confusing that the objectives section includes a discussion of both the aims of the wider study and this paper. It might be better to frame this section around the need to improve IYCF practices, the potential effectiveness of school based interventions, and only discuss the aims of this paper. In the design section, readability could be improved if the focus was only on this paper. In the results section, include a more comprehensive summary of the findings. Introduction The introduction frames this paper around the importance of improving IYCF within children 6-24 months old. However, the intervention targets school aged children. This discrepancy is not sufficiently addressed within the paper. It may be helpful to include additional detail about the setting (i.e. many households with children between 6-24 months also have school aged children) or explicitly state potential mechanisms for impact. Making the introduction more concise could improve readability. In particular, first, it may be beneficial to minimise discussions about the PANChSHEEEL project in the introduction. This can be detailed in the methodology. Second, paragraphs two and three on page 5 of the introduction could be more concise. Detailed descriptions about school based programmes may be better suited for the discussion. Line 30 (Page 5, Introduction) – possible error “as long”, meant “as well” Line 45 (Page 5, Introduction) – detail the target population Methodology Insert a few sentences at the beginning of the methodology to state that this paper is part of a wider study.
---

	At the end of the first paragraph, insert a sentence stating which phase of the wider study is detailed in this paper. Provide reflexivity details for the female community researchers who collected data. Detail why a distinction is made between paternal and maternal grandmothers, and why the focus is on paternal grandmothers. In the data collection section, mention that interviews were recorded with an audio device and mention that field notes were taken during the interviews. In the data management section, briefly detail the IYCF themes for each intervention point. It is not clear who analysed the qualitative data. Was it the entire research team? If so, how were the findings consolidated? Findings Line 26 (Page 1, Findings) - social mapping details etc should be included in the methodology. In order to improve the readability of the findings, make them more cohesive, and highlight the link between the findings and the paper aims, it may be helpful to reorganise the findings into headings such as challenges, drivers, and opportunities. Second, it may be helpful to have the first sentence of each subsection explicitly linking the findings to the research aims. Discussion Most of the information about literature examining school based interventions in the introduction can be moved to the discussion. This would strengthen the discussion and present the findings of this paper within the context of existing literature. Beyond the importance of disseminating nutrition education and the receptibility of such information among primary caretakers, it would also be beneficial to discuss (even briefly) the importance of primary caretakers' ability to implement recommended practices.
--	---

REVIEWER	Peter Rohloff Brigham & Women's Hospital and Children's Hospital
REVIEW RETURNED	13-Apr-2021

GENERAL COMMENTS	This is a really interesting initiative that is being conducted by a large and thoughtful team. I enjoyed reading it and I learned a lot from the paper. That being said, I think that there some serious issues with framing and with complexity that need to be addressed, I think through a fairly complete restructuring of the paper and so I would recommending declining the paper in its current form. I do not think this reflects on the quality of the work, but the presentation needs rethinking. First, this is part of a large and complex long-term study. However, what is presented here is really very straightforward, and it suffers from an overly long description of the longer term study design and methods. I recommend eliminating all of this, except for a very brief framing section describing the overall goals of the team's work, and focusing on the fact that what is presented here is just a very simple qualitative study of the school environment. Second, the author's frame their study in terms of HEEE. This term/theory is not well described, and I'm not familiar with it. A search of google doesn't turn up any usages, and a search of
--

	Pubmed reveals only one other study publication by the same team. Therefore, this theoretical framework must be described and characterized. What does “engineering” mean in this context? Does this mean the built school environment? All of this is fairly unclear. Third, although the overall framework of the study is improving infant/toddler health (IYCF practices) and although admittedly there is some evidence that school age children can bring home lessons on nutrition, etc. that can be incorporated by the family, the paper actually only presents very straight-forward results about school attendance, parental participation, school environment, school lunches. So this is really mostly a study of the school environment and the thematic content tenuously engages with IYCF practices and the feasibility of IYCF themes delivered in schools (to whom? parents or children?) being disseminated back to homes. I understand this is the overall trajectory of the team’s work, but that data is not evident here and reframing this paper to focus more on what is actually explored would be most appropriate I think. Fourth, although I appreciate the attention given to the CORE-Q checklist, there is some disconnect between what is described and what is presented here. For example, “focus was on collecting data on the infant and young child feeding practices” even though no data is given on IYCF practices in this paper. Again, this seems like a general methodological description of the entire team’s work, rather than a description of what is actually presented here. Additional needed details include how needs sample size was estimated and what measures were taken to assess thematic saturation. Details on coding and analysis are also lacking (were data collected and coded in Wagdi or another language? Who did the coding and developed themes? Where data coded by multiple coders? How were disagreements reconciled?). Additionally it is not completely clear what the role of the community researchers vs community champions was in data collection and whether they had a role in data analysis. I do hope that these comments are taken as constructive, and are helpful to the authors in reaching a wide audience with their work.
--	---

VERSION 1 – AUTHOR RESPONSE

Reviewer: 1

Ms. Nakawala Lufumpa, University of Birmingham College of Medical and Dental Sciences

Comments to the Author:

This is important work and a very good study. Please find attached a document which details my comments and recommendations to strengthen the paper.

****Please see attached comments ****

It is a bit confusing that the objectives section includes a discussion of both the aims of the wider study and this paper. It might be better to frame this section around the need to improve IYCF practices, the potential effectiveness of school-based interventions, and only discuss the aims of this paper.

In the design section, readability could be improved if the focus was only on this paper. In the results section, include a more comprehensive summary of the findings.

For clarity purposes, we have addressed all of the aforementioned comments in the relevant section, i.e. Objectives, Design and Results section, of our abstract. We thank you for your detailed input.

The introduction frames this paper around the importance of improving IYCF within children 6-24 months old. However, the intervention targets school aged children. This discrepancy is not sufficiently addressed within the paper. It may be helpful to include additional detail about the setting (i.e. many households with children between 6-24 months also have school aged children) or explicitly state potential mechanisms for impact.

Thank you - In response to this comment, we have shortened the section on the importance of improving IYCF practices 6-24month olds, and put the primary focus on interventions in school-aged children / the school as a setting for community intervention.

Making the introduction more concise could improve readability. In particular, first, it may be beneficial to minimise discussions about the PANCHSHEEL project in the introduction. This can be detailed in the methodology. Second, paragraphs two and three on page 5 of the introduction could be more concise. Detailed descriptions about school-based programmes may be better suited for the discussion.

As mentioned, we have shortened the Introduction significantly and have shifted the explanation on the Panchsheel initiative as well as descriptions about other school-based programmes into the Methodology and Discussion section respectively.

Line 30 (Page 5, Introduction) – possible error “as long”, meant “as well” Line 45 (Page 5, Introduction) – detail the target population

Thank you – the incorrect term has been replaced.

Insert a few sentences at the beginning of the methodology to state that this paper is part of a wider study. At the end of the first paragraph, insert a sentence stating which phase of the wider study is detailed in this paper.

Both points have been addressed in the first paragraph. As aforementioned, the amount of information on the wider study has been reduced in the Introduction, and instead been incorporated in the Methodology section. Thank you.

Provide reflexivity details for the female community researchers who collected data. Detail why a distinction is made between paternal and maternal grandmothers, and why the focus is on paternal grandmothers.

The rationale behind this distinction is now outlined in our manuscript for clarity purposes. Please refer to the 'Data collection' section to see our tracked changes in response to this comment.

In the data collection section, mention that interviews were recorded with an audio device and mention that field notes were taken during the interviews. In the data management section, briefly detail the IYCF themes for each intervention point. It is not clear who analysed the qualitative data Was it the entire research team? If so, how were the findings consolidated?

Thank you – Information to address all the aforementioned points has been incorporated in the 'Data collection' and 'Data Management' section in our manuscript.

Line 26 (Page 1, Findings) - social mapping details etc should be included in the methodology.

We have adjusted the manuscript accordingly. This can be found in the 'Study design' subsection of our Methodology.

In order to improve the readability of the findings, make them more cohesive, and highlight the link between the findings and the paper aims, it may be helpful to reorganise the findings into headings such as challenges, drivers, and opportunities. Second, it may be helpful to have the first sentence of each subsection explicitly linking the findings to the research aims.

Thank you. The results / findings section of our paper has been reorganised into the subheadings 'challenges, drivers, and opportunities'. Our figures have been revised accordingly. We have attempted to clarify the link between each of our findings and our research aim.

Most of the information about literature examining school-based interventions in the introduction can be moved to the discussion. This would strengthen the discussion and present the findings of this paper within the context of existing literature.

As previously mentioned, the description of other school-based interventions has been moved into the discussion section.

Beyond the importance of disseminating nutrition education and the receptibility of such information among primary caretakers, it would also be beneficial to discuss (even briefly) the importance of primary caretakers' ability to implement recommended practices.

Thank you – we have incorporated some information on the implementation of nutrition education, through the co-location with the already established Angawadi centres.

Reviewer: 2

Dr. Peter Rohloff, Brigham & Women's Hospital and Children's Hospital

Comments to the Author:

This is a really interesting initiative that is being conducted by a large and thoughtful team. I enjoyed reading it and I learned a lot from the paper.

That being said, I think that there some serious issues with framing and with complexity that need to be addressed, I think through a fairly complete restructuring of the paper and so I would recommending declining the paper in its current form. I do not think this reflects on the quality of the work, but the presentation needs rethinking.

First, this is part of a large and complex long-term study. However, what is presented here is really very straightforward, and it suffers from an overly long description of the longer term study design and methods. I recommend eliminating all of this, except for a very brief framing section describing the overall goals of the team's work, and focusing on the fact that what is presented here is just a very simple qualitative study of the school environment.

Thank you. In response to the comments from Reviewer 1 and 2, we have significantly re-structured our paper and placed our primary focus on the qualitative study presented, rather than the wider PANChSHEEEL initiative. Only a brief framing section has been kept in the paper's methodology section.

Second, the author's frame their study in terms of HEEE. This term/theory is not well described, and I'm not familiar with it. A search of google doesn't turn up any usages, and a search of Pubmed reveals only one other study publication by the same team. Therefore, this theoretical framework must be described and characterized. What does "engineering" mean in this context? Does this mean the built school environment? All of this is fairly unclear.

We have referenced further papers referring to the concept of HEEE for clarity purposes. We thank you for your input.

Third, although the overall framework of the study is improving infant/toddler health (IYCF practices) and although admittedly there is some evidence that school age children can bring home lessons on nutrition, etc. that can be incorporated by the family, the paper actually only presents very straight-

forward results about school attendance, parental participation, school environment, school lunches. So this is really mostly a study of the school environment and the thematic content tenuously engages with IYCF practices and the feasibility of IYCF themes delivered in schools (to whom? parents or children?) being disseminated back to homes. I understand this is the overall trajectory of the team's work, but that data is not evident here and reframing this paper to focus more on what is actually explored would be most appropriate I think.

As aforementioned, this comment has been addressed through a restructuring / refocus of our manuscript. We hope our amendments meet the reviewer's expectations. Thank you.

Fourth, although I appreciate the attention given to the CORE-Q checklist, there is some disconnect between what is described and what is presented here. For example, "focus was on collecting data on the infant and young child feeding practices" even though no data is given on IYCF practices in this paper. Again, this seems like a general methodological description of the entire team's work, rather than a description of what is actually presented here.

Thank you – The CORE-Q list has been updated following changes in our manuscript.

Additional needed details include how needs sample size was estimated and what measures were taken to assess thematic saturation. Details on coding and analysis are also lacking (were data collected and coded in Wagdi or another language? Who did the coding and developed themes? Where data coded by multiple coders? How were disagreements reconciled?). Additionally it is not completely clear what the role of the community researchers vs community champions was in data collection and whether they had a role in data analysis.

We appreciate your input. The aforementioned questions on the coding and analysis of our results have been answered in the Data collection and Data Management sections of our revised manuscript. We have further clarified and distinguished the roles of the different collaborators, e.g. community researchers vs. community champions.

I do hope that these comments are taken as constructive, and are helpful to the authors in reaching a wide audience with their work.

VERSION 2 – REVIEW

REVIEWER	Peter Rohloff Brigham & Women's Hospital and Children's Hospital
REVIEW RETURNED	03-Nov-2021
GENERAL COMMENTS	Authors have done a lot of work and the paper is much better. It should be published.